# South African university students' experiences of online group cognitive behavioural therapy: Implications for delivering digital mental health interventions to young people

Xanthe Hunt[1] (ID), Dionne C. Jivan[2], John A. Naslund[3], Elsie Breet[1] and Jason Bantjes[1,4]

[1]Institute for Life Course Health Research, Department of Global Health, Faculty of Medicine and Health Sciences, Stellenbosch University, Cape Town, South Africa; [2]Department of Psychology, Faculty of Arts and Social Sciences, Stellenbosch University, Stellenbosch, South Africa; [3]Department of Global Health and Social Medicine, Harvard Medical School, Boston, MA, USA and [4]Alcohol, Tobacco and Other Drugs Research Unit, South African Medical Research Council, Cape Town, South Africa

## Research Article

**Keywords:**
digital mental health; cognitive behavioural therapy; university students; mHealth; group therapy

**Corresponding author:**
Xanthe Hunt;
Email: xanthe@sun.ac.za

## Abstract

Mental disorders are common among university students. In the face of a large treatment gap, resource constraints and low uptake of traditional in-person psychotherapy services by students, there has been interest in the role that digital mental health solutions could play in meeting students' mental health needs. This study is a cross-sectional, qualitative inquiry into university students' experiences of an online group cognitive behavioural therapy (GCBT) intervention. A total of 125 respondents who had participated in an online GCBT intervention completed a qualitative questionnaire, and 12 participated in in-depth interviews. The findings provide insights into how the context in which the intervention took place, students' need for and expectations about the intervention; and the online format impacted their engagement and perception of its utility. The findings of this study also suggest that, while online GCBT can capitalise on some of the strengths of both digital and in-person approaches to mental health programming, it also suffers from some of the weaknesses of both digital delivery and those associated with in-person therapies.

## Impact statement

This study found that online group therapy for university students can capitalise on some of the strengths of both digital and in-person approaches to mental health programming, being easily accessible (akin to many digital interventions) and allowing for interpersonal connection (akin to many in-person therapies). However, we also found that online therapy suffers from some of the weaknesses of both digital delivery and those associated with in-person therapies. For instance, because digital interventions are designed to be scalable, they are often manualised. However, manualisation made some users feel that the programme lacked personalisation, and flexibility and responsiveness in the here-and-now. Other weaknesses of the digital platform included the lack of accountability and difficulty managing group dynamics online. The implications of this study are that questions still remain about whether – from an implementation perspective – it is more useful to think of online therapies as a digital intervention (like an app) or simply as group therapy that happens to be held on a digital platform (like telepsychiatry). Many of the issues raised in this study are ones germane to the literature on mental health apps in low- and middle-income countries, including convenience and personalisation of scalable interventions. However, our findings show that the relational elements of the intervention – the 'human' elements – are important to participants, and to users' sense of the programme as 'real' (rather than virtual). Difficulties arise, however, because precisely the factors which give the programme its 'real' feel for participants (synchronous delivery, the requirement of a clinician to deliver content, and need for strong Internet bandwidth among users), are those which pose barriers to scale.

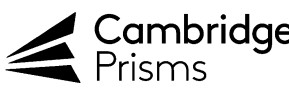



## Introduction

Mental disorders are common among university students (hereafter 'students') (Auerbach et al., 2018; Bantjes et al., 2019). A recent survey of first-year students across eight countries reported that the lifetime prevalence of common mental disorders was 35.3% and the 12-month prevalence was 31.4% (Auerbach et al., 2018). Major depressive disorder (MDD) and generalised

anxiety disorder (GAD) are the most common conditions, with 12-month prevalence rates of 18.5 and 16.7%, respectively (Auerbach et al., 2018). Left untreated, mental disorders pose serious risks to students' educational achievement and functioning (Alonso et al., 2019). Compared to their peers, students with mental disorders are less likely to cope with the transition to university (Al-Qaisy, 2011; Grøtan et al., 2019), have lower academic achievement (Grøtan et al., 2019) and worse long-term employment, productivity, relationships and health outcomes (Eisenberg et al., 2012). Mental health concerns among students have been exacerbated over the course of the COVID-19 pandemic, as reflected in numerous studies from diverse settings reporting a worsening of mood, greater perceived stress and increased alcohol consumption during this period (Charles et al., 2021; Copeland et al., 2021; Visser and Law-van Wyk, 2021).

Yet, despite the high burden of need and the risks of foregoing treatment, many students with mental disorders do not receive any treatment at all (Eisenberg et al., 2012; Bruffaerts et al., 2019). Reasons for the treatment gap include stigma, a lack of awareness of their need for care, and low knowledge and/or acceptability of available resources (Ibrahim et al., 2019; Hilliard et al., 2022). For young people specifically, the desire to deal with challenges alone, beliefs about the ineffectiveness of therapy, and competing priorities and demands on their time are key barriers to seeking care (Ennis et al., 2019). Resource constraints in low- and middle-income countries (LMICs) exacerbate the treatment gap on university campuses as mental health services are often under-resourced, over-stretched, or entirely absent (Demyttenaere, 2004; Saxena et al., 2007; Docrat et al., 2019). Furthermore, traditional in-person psychotherapeutic treatment options are often not feasible, affordable or easily scalable to address the large need for care. And, given low rates of engagement in services (Vanheusden et al., 2008; Bantjes et al., 2020), it is also possible these modalities may not be the optimal approach to reach all (if not the majority) of students.

In the face of these circumstances, there has been interest in the role that digital mental health solutions could play in meeting youth mental health needs, particularly in resource-constrained settings (Waegemann, 2010; Grist et al., 2017; Punukollu and Marques, 2019; Leech et al., 2021; Bantjes et al., 2022). The COVID-19 pandemic accelerated the utilisation of digital interventions to provide remote psychiatric care, helping to establish digital technologies as viable treatments (Stein et al., 2022a). There has been a proliferation of digital interventions brought to market in the past decade (Lehtimaki et al., 2021).

However, a recent systematic review and meta-analysis demonstrated mixed effectiveness of these interventions and experts have cautioned against unrealistic optimism about digital mental health interventions' potential to address the broad need for mental health services or supplant in-person services (Grist et al., 2017). Evidence of implementation challenges facing digital interventions broadly (Ford II et al., 2015; van Olmen et al., 2020) has highlighted the need for consistent demonstrations of their usability and effectiveness (Lehtimaki et al., 2021; Stein et al., 2022b). Moreover, researchers and practitioners have noted context-specific challenges facing users in LMICs, including practical barriers such as the cost of cell phone data, lack of connectivity and limited access to smart devices (van Olmen et al., 2020). Despite increasing access to digital devices across most contexts, understanding the barriers to using digital mental health solutions in LMICs is essential to close the persisting digital divide (Bantjes, 2022).

Digital interventions exist at a range of intensities and levels of digitization: Some, like online therapy, have digital delivery, but otherwise resemble traditional, in-person mental health services, as reflected by the synchronous connection to a mental health provider. Others, like apps, are fully digitised and exist at high (e.g. artificial intelligence chatbots) and low (e.g. mood monitoring calendars) intensities. Digital interventions can be conceptualised as existing along these two dimensions (digitization and intensity) as per Figure 1. There are of course other dimensions along which digital interventions can be conceptualised, including introducing the necessity for a practitioner, the potential to scale-up interventions, and the degree to which individuals can choose how and when to engage with the interventions.

Different types of digital mental health interventions have specific strengths, such as being able to scale to many individuals at low cost or being easily accessible at the time and place of an individual's choosing. However, weaknesses are also notable, such as the requirement for continuous data access and challenges sustaining user engagement (Martinez-Martin and Kreitmair, 2018).

In an attempt to address the twin challenges of limited scale of in-person psychotherapy, and the criticisms levelled against fully digitised interventions such as apps, a group of researchers from South Africa (SA) and the United States (US) developed an online group cognitive behavioural therapy (GCBT) intervention. The programme, designed for delivery via videoconferencing to students with mental health problems, aimed to capitalise on the cost- and scale benefits of a digital platform, but not lose the effectiveness and acceptability of in-person therapy. The programme was piloted in 2020 by Bantjes et al. (2021) and found to show promise as an effective and sustainable intervention for the treatment of anxiety and depression among students.

However, the pilot results do not allow for a nuanced understanding of how the intervention was experienced by students, including important questions around whether this 'hybrid' format – of a traditional therapy modality delivered on a digital platform – was acceptable to the programme's users. Moreover, recent reviews have called for additional research into online group therapy, particularly given the paucity of evidence on the implementation of these programs (Weinberg, 2020). The objective of this study was to conduct qualitative interviews with participants from the 2020 pilot study to explore their perspectives about the intervention's

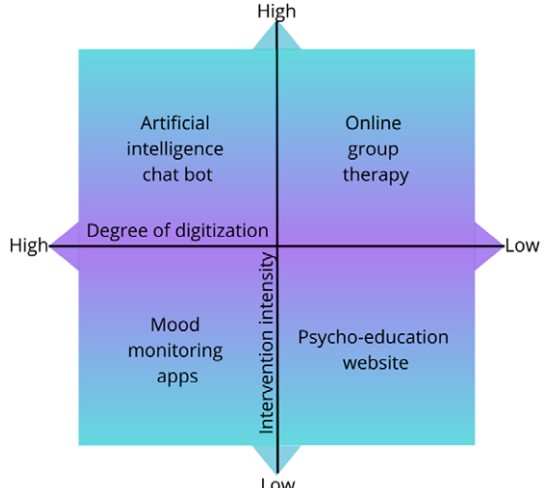

**Figure 1.** Digital interventions as organized along the dimensions of digitization and intensity.

content and delivery, and to provide insights to guide future implementation of the GCBT intervention.

## Methods

### Design

This study is a cross-sectional, qualitative inquiry into students' experiences of an online GCBT intervention. It is a sub-study of a pilot open-label trial of the GCBT intervention in which 158 students were enrolled into the GCBT intervention (Bantjes et al., 2021). Qualitative data were collected at two points during the study, firstly through a single open-ended online questionnaire item sent to all 158 participants, and secondly through in-depth interviews with 12 randomly selected participants. While the correct processes for determining sample size in qualitative studies are debated, we used the guidelines from Clarke and Braun (2013).

### Setting

Students for this intervention were recruited from a university in SA, a country with high rates of mental disorders and low treatment rates (Bruwer et al., 2011; Newson et al., 2021). Mental disorders are particularly common among young people with a recent study reporting the 12-month prevalence rate of common mental disorders to be 31.5% among first-year students, with MDD and GAD being the most common disorders (12-month prevalence of 13.6 and 20.8%, respectively)) (Bantjes et al., 2019). In SA, roughly 92% of individuals in need of mental health services do not have access to them (Docrat et al., 2019). Moreover, where resources are available, utilisation rates are often low (Bantjes et al., 2020). In this study, no symptom threshold was set on participation, and so participants included both individuals with and without clinically significant symptoms of depression and/or anxiety.

### Participants and procedure

Recruitment for the pilot open-label trial is described in depth in Bantjes et al. (2021) but, briefly, involved information about the intervention being posted once on a student affairs Facebook page at a university in South Africa in mid-2020. The post explained that web-based groups were being offered to help students learn psychological skills to reduce symptoms of anxiety and depression. The 175 students who responded within 24 h to the notice provided informed consent and completed a baseline assessment before being randomised to one of 15 GCBT groups. No symptom threshold was placed on participants, and anyone who wanted to participate was eligible to do so. For the qualitative sub-study, which is the subject of this article, data were collected in two formats. First, in September 2020, all 158 students who participated in the GCBT intervention were invited via an online questionnaire to give qualitative feedback about the online intervention. Then, a random sample of participants was invited to attend semi-structured in-depth interviews. Five rounds of recruitment emails were sent to 12 participants each time (60 in total). In each recruitment batch, the 12 participant email addresses were randomly selected from the total sample using a random number generator. One reminder email was sent to each participant before a new batch of emails was sent out. The process of sending recruitment emails was continued until the sample size for the in-depth interviews had been achieved (n = 12). These in-depth interview participants were then interviewed via MS Teams by a trainee clinical psychologist.

These interviews were recorded, transcribed and analysed. Ethical approval was obtained from the Psychology Ethics Committee, Stellenbosch University, (N19/10/145, Project ID: 12977). All participants provided informed consent.

### Instruments

The online questionnaire entailed a single open-ended field which prompted participants to insert text in response to the question, "*Please use the space below to give us any other feedback about your experience of the group. Tell us what you liked or did not like, and what you think we can do to make this group more helpful to other students in the future*". For the in-depth interviews, a semi-structured interview guide was used which contained open-ended questions relating to students' experiences of the online GCBT intervention. These questions were guided by our desire to understand the following:

1. Acceptability of the 'hybrid' format – of a traditional therapy modality delivered on a digital platform
2. Barriers and facilitators to engagement with the psychological skills training group intervention
3. Reflections on the digital intervention as it compares to any other experiences with mental health treatments
4. Contextual and cultural appropriateness of the intervention and its method of delivery

The semi-structured interview guide for the individual interviews can be found in Supplementary File 1.

### Details of the GCBT intervention

The online GCBT intervention, delivered via MS Teams, consisted of 10 weekly group sessions which were 60–75 min each. The content of the intervention is described in detail in Bantjes et al. (2021), but session topics included emotional triggers and automatic thoughts, identifying emotional triggers, challenging automatic thoughts and core beliefs, recognising stressors and using strategies to solve interpersonal and emotional problems, overcoming rumination and guilt, behavioural activation and coping with difficult emotions. Each group had between 10–12 members who attended 10 workshops organised into 5 topics. The membership of groups was largely fixed. Each participant got an interactive workbook which served as a guide and included activities and summaries focusing on the main ideas and skills for each session.

### Data analysis

For analysis, the data from the survey and the in-depth interviews were coded separately. This was because, at the time of analysis, the survey data were seen as routine monitoring and evaluation data, while the in-depth interviews were designed as a qualitative sub-study. Data from each dataset were anonymised and pseudonyms were assigned. They were then analysed via inductive thematic analysis and the six-phase approach outlined by Braun and Clarke (2006). The phases of this approach include familiarisation, coding, generating themes, reviewing themes, defining and naming themes and writing up (Braun and Clarke, 2006). All data were managed and analysed in Microsoft Excel (Microsoft Corporation, 2018), where units of meaning from the survey and qualitative transcripts were pasted into a spreadsheet, and a code was assigned to each new unit of meaning using adjacent columns. Units of meaning – codes – were then organised together into larger, descriptive groups, and

names of these groups (themes) were assigned. Once the analysis was complete, it became apparent that the prompt for the survey data collection had generated some unique responses which were not captured in the in-depth interviews. The decision was then taken to include both sets of data – from the online questionnaire and the in-depth interviews – in one final analysis. Data were combined and results were written up. All analyses were conducted by two independent coders: E.B. and D.C.J. for the survey data and X.H. and D.C.J. for the in-depth interview data. All analyses were reviewed for quality control by J.B.

## Results

A total of 125 respondents completed the online questionnaire (out of $N = 158$; 79% response rate), and 12 of the 125 were randomly selected to participate in in-depth interviews. Eighty-six percent of the questionnaire respondents self-identified as female and the mean age of the sample was 21.96 years (see Table 1 for detail). The demographic profile of the respondents to the in-depth interviews was broadly similar to those of the larger sample (70% self-identified as female, and their mean age was 21.01 years).

The findings from both the online questionnaire and the in-depth interviews provide insights into how the context in which the intervention took place, students' need for and expectations about the intervention; and the online format impacted their engagement and perception of its utility. The themes identified in the data are summarised in Table 2. The themes arising primarily from the online questionnaire are flagged in italics.

### *Opportunities for connection and continuity*

The online intervention was delivered in 2020 during the first wave of the COVID-19 pandemic in SA. Participants reflected on this, noting that the pandemic and the measures required for its containment created a greater need for mental health services. Many participants noted that the intervention provided them with a much-needed therapeutic space for their mental health difficulties, as well as space to learn skills to deal effectively with the new stresses associated with the pandemic. Furthermore, the weekly group sessions provided opportunities for interacting with other students and created a sense of community. One participant explained:

> There was some really hard-core isolation going on and the group was a nice way of hearing people's voices … There was a cage effect, we were stuck not knowing what to do and were stuck with our own thoughts and in that way the content was good coz it gave us a structure of how to proceed going through these thoughts… I think a lot of people also felt that way, they used the group as a social platform as well.
> – May, 25-year-old female, in-depth interview

**Table 2.** Themes

| Themes | Opportunities for connection and continuity | Reality versus expectations | Group format and online spaces as barriers and facilitators | Perception of therapeutic value of the intervention |
| --- | --- | --- | --- | --- |

Many participants said the intervention served as a stabilising force in their lives, providing routine and predictability at a time of instability. As one participant noted:

> I'm a person that likes to have things to do and likes processes, rules and methods. For me I found [the group] very helpful and reassuring… particularly in this context of Corona.
> – James, 19-year-old male, in-depth interview

While many students felt that the context of the pandemic amplified the need for an online mental health intervention, others reflected on the way in which the reality of the pandemic and the broader socio-political context of the country butted up against the very pragmatic approach of CBT. One participant, for instance, felt that some of the content delivered through the intervention was not particularly sensitive to current realities or socio-political issues. She noted:

> It was only through the check-ins and check-outs and the little bit of personal connection that we touched on the fact of COVID. I thought maybe to make it more practical, for example, this year they could have done a little bit more. Like, how do you then for example, do SMART goals in a situation of a pandemic or in the previous years where there were major student protests.
> – Nomanono, 22-year-old female, in-depth interview

### *Reality versus expectations*

In general, participants had low initial expectations for the online GCBT intervention. Since it was called a 'psychological skills group' in the advertising material circulated on-campus social media, participants appeared to expect a psychoeducational programme rather than therapy. One respondent described her experience, saying:

> Initially my impression of it was, it was going to be a course, and I am generally interested in CBT and all things psychology.
> – Sarabi, 22-year-old female, in-depth interview

Other participants had understood that the intervention was going to be group therapy, but admitted that they did not expect it to have any impact on their mental health. Despite low expectations, many participants noted that the intervention was surprisingly helpful and engaging. One young man explained:

> I saw an email and I didn't really expect much from it, I didn't think it would be helpful because I was like okay, I am in need of some form of you know, we were in lockdown and not on campus

**Table 1.** Participant demographics for the whole sample

| | Followed-up (*n* = 125) | | | Lost to follow-up (*n* = 33) | | | |
| --- | --- | --- | --- | --- | --- | --- | --- |
| | Mean | SE | SD | Mean | SE | SD | *p*-value |
| Female | 85.6% | 3.2% | 35.3% | 84.8% | 6.3% | 36.4% | .86 |
| Age | 22.0 | 0.4 | 4.5 | 23.5 | 1.4 | 7.9 | .29 |
| Undergraduates | 78.4% | 3.7% | 41.3% | 66.7% | 8.3% | 47.9% | .16 |
| Number of sessions attended | 6.8 | 0.2 | 2.6 | 4.8 | 0.6 | 3.2 | **.00** |

*Note*: The value in bold is significant at the <0.05 level.

anymore, that sort of thing. So, I didn't expect it to be as helpful and such a nice safe environment as it was.
– Isaac, 23-year-old male, in-depth interview

### Group format and online spaces as barriers and facilitators

Many participants found the online format of the group helped create a 'safe space' for self-disclosure and self-discovery, without the anxiety of direct face-to-face contact with others. This was particularly salient for participants with social anxiety, as exemplified in the following participant's account:

I'm a relatively shy person and being online gives you a bit of confidence because there is a barrier between you and other people to an extent, you kind of feel a little braver and sharing.
– Kavitha, 20-year-old female, in-depth interview

In contrast, some participants experienced the online environment as a barrier to creating and maintaining interpersonal connections. One young man noted:

I think the fact that it is online is just kind of limited, so I think they did all they could to make it very, uhm to connect us well on the medium that was used… I think the human connection was lacking.
– Isaac, 23-year-old male, in-depth interview

Participants spoke about the lack of visual cues as a barrier to their engagement and an impediment to the therapeutic process, as one participant explained:

A lot of things seemed quite distant, like you're still working through a computer and like with the screen when it came to like really practical like almost role play stuff it was difficult.
– Simon, 19-year-old male, in-depth interview

The online delivery of the intervention helped to make the intervention convenient compared to traditional in-person services. Participants said that this ease of use facilitated attendance and engagement. As one participant explained:

When you are online … then it's easier to plan around your schedule because sometimes it's difficult for everyone to meet in one place for an hour like that, like once a week.
– Tracey, 19-year-old female, in-depth interview

However, as much as convenience was a strength in many participants' eyes, some individuals also noted that because they did not have to invest much effort in attending, they also easily forgot to join. This was also tied to the fact that the program was delivered online. As one respondent noted:

I felt a lot more accountable when I had to attend in person meetings because … it's just a lot easier to forget an online meeting than to forget an in-person meeting.
– Gina, 22-year-old female, in-depth interview

Some participants also found themselves dividing their attention between the intervention and other internet media:

I could check Twitter or I could just reply to a WhatsApp or like you know there are 1000 other things. I am sitting in the comfort of my own home and there are so many things that distract you, you know so it's really hard to … focus.
– Alex, 20-year-old female, in-depth interview

Many participants mentioned that the group provided a sense of community and cohesiveness that validated their own experiences and feelings. One respondent reflected on this, saying:

I think it's the fact that you are not alone in what you're going through. That kind of thing was quite eye opening to me is you

always think that, you know, you're facing this whole thing on your own.
– Gina, 22-year-old female, in-depth interview

Participants spoke specifically about the intervention helping normalise their experiences by allowing them to see the similarities between their own struggles and those of other students. Alice explained:

Even though they are going through very different things than any of us were, we had the same sort of things that were triggering our emotions or the reasons for us being there.
– James, 19-year-old male, in-depth interview

While most students were pleasantly surprised by the programme and found the content useful, there were some who spoke of not having an entirely positive experience. For instance, one participant described how her individual needs were not met due to the inability of the facilitators to manage individual versus group needs:

[The facilitators] didn't make anyone speak but then they also didn't allow everyone to speak … the one lady she almost at times would have her own one on one counselling therapy and we would just be spectators and she would happily talk on for 20 minutes.
– Nomanono, 22-year-old female, in-depth interview

### Perception of therapeutic value of the intervention

Participants reported that skills-based group therapy which included a workbook was very helpful in their own learning and growth over 10 weeks. Although some mentioned that they may have preferred a more relaxed group to talk to people in general, they still benefitted from the skills even when not directly applicable to them. One respondent said:

I really liked the CBT approach to therapy and kind of helping to retrain your thoughts. It was very insightful and I learned a lot, I think the content was really great, very educational, easy to follow and well set out.
– Isaac, 23-year-old male, in-depth interview

Participants reported that they appreciated having the workbooks and found the practical activities in these helpful. One participant affirmed this saying:

The worksheets that students received after a group session were very helpful because students could always go through what was discussed during the group session using the worksheets later on.
– Mihlali, 17-year-old female, online questionnaire

The content of the workbooks served as a summary of the skills covered in the sessions, but also provided participants with opportunities to continue their engagement with the content in-between sessions. However, not all the participants found the structured worksheets and workbook helpful. One participant, for example, said:

I thought this group could be a place where we could just share and talk through our stuff, but there was a schedule and activities and stuff to read through and actually it just felt like it was contributing to my workload.
– Katie, 19-year-old female, online questionnaire

While many participants appreciated the content and skills-based approach of the intervention, many of them also expressed a wish for a less structured and less content-driven and directive approach. They had an unmet expectation that the groups would provide more space for personal disclosure, discussion and interaction with

other participants. Having time and opportunities to interact with other students is a valued component of the intervention, as shown in the following feedback from one participant:

> I enjoyed the interaction within the group and hearing people's opinions or how they deal with certain situations.
> – Shanice, 22-year-old female, online questionnaire

Finally, participants attributed the success of the intervention in large part to the skill and attitude of the facilitators, saying things like:

> I felt that facilitators were open minded, encouraging and empathic and also knew a lot.
> – Jack, 27-year-old male, online questionnaire

It was evident that most participants felt a personal connection to the facilitators and perceived them to be warm, welcoming, non-judgmental, skilful and knowledgeable. These positive feelings towards the facilitators seemed to promote participants' engagement in the process and enabled them to receive the skills being offered. One respondent articulated this by saying:

> It was a great experience and a safe environment where I could share my feelings knowing I would not be judged.
> – Sabrina, 18-year-old female, online questionnaire

The feeling of safety and the non-judgmental environment created by the facilitators appears to have been an integral component of the success of the intervention.

## Discussion

Students who participated in a 10-week online GCBT intervention delivered on university campus in South Africa largely found the intervention engaging and helpful. However, our findings highlight several key considerations for implementing these kinds of interventions with young people. These considerations are particularly important to address if such programmes are going to fulfil their potential to lessen the mental health treatment gap among university students in LMICs.

As noted in the introduction to this article, our team developed and tested an online GCBT programme as an intervention which took a middle road between digital and in-person services. Our findings suggest that, while the programme capitalises on some of the strengths of each approach, it also suffers from some of the weaknesses of both digital delivery and those associated with in-person therapies. Moreover, it appears that some of the features of digital delivery mean that those strengths which are associated with in-person therapies are diluted.

We also found that context plays a central role in determining how and for whom and when digital interventions work. The pandemic made online delivery of the intervention facilitative of engagement and students' busy schedules meant that the flexibility afforded by the online group was appreciated. However, when it came to intervention content (as opposed to delivery), context posed a challenge: Because digital interventions are designed to be scalable, they are often manualised. However, manualisation made some students feel that the programme lacked personalisation. An important direction for future work refining scalable digital interventions will be to understand how to optimise opportunities to ensure that a programme has sufficient fidelity to the evidence-based treatment manual whilst also being able to respond to the needs and priorities of students. Flexibility and responsiveness in the here-and-now are potentially among the biggest

strengths of online group interventions, compared to other digital interventions (like apps) where the content is often fixed. Finding ways to harness the flexibility of synchronous online group therapies while retaining fidelity to the core CBT skills which make up this online intervention is integral to maximising the benefits of this intervention.

Interestingly, most participants felt that the intervention exceeded their expectations. On the surface of things, this is a positive outcome for the intervention pilot. However, part of the reason that their expectations had been exceeded was because they thought that they were signing up for psychoeducation rather than therapy. This highlights the need to develop marketing messages that are better tailored to the target population of students, as well as to manage students' expectations when recruiting them to digital interventions, a point that has been made by other authors (Gericke et al., 2021). Expectations about psychotherapy are an important determinant of treatment outcome (Greenberg et al., 2006), so managing students' expectations of digital intervention is important.

Other weaknesses of online GCBT included the lack of accountability (students could easily forget about or miss the sessions), and the limited opportunities for interpersonal connection. Many initiatives in mental health are examining how peer support can be delivered online (Melling and Houguet-Pincham, 2011; Ali et al., 2015). Some of the lessons from this literature might well be used to improve the peer-to-peer engagement aspect of synchronous therapeutic groups. Developing approaches to improve accountability, however, require additional research.

Relatedly, while the group setting seemed to have offered some participants a sense of kinship with their peers, which made them feel at ease, others felt that the group dynamics had not 'worked' online. Management of group dynamics in therapy is a well-established area of study in traditional psychotherapy (Bion, 1952; Sutherland, 1985; Scheidlinger, 1997), and is receiving increasing attention for online groups (Biagianti et al., 2018; Weinberg, 2020, 2021). Emerging studies have also found that back-and-forth interaction between peers within online platforms appears to promote retention (Sharma et al., 2020). For digital interventions to be effective, careful consideration needs to be given to how peer-to-peer interaction is facilitated. In online GCBT, this could be achieved, for example, by using break-out rooms on videoconferencing platforms.

At the start of this article, we introduced the idea that digital interventions exist at a range of intensities and levels of digitization, and that there are strengths and weaknesses afforded by different degrees of digitization and intensity. One of the questions raised by this study is whether – from an implementation perspective – it is more useful to think of online GCBT as a digital intervention (akin to an app) or simply as group therapy that happens to be held on a digital platform (akin to telepsychiatry). Many of the issues raised in this study are ones germane to the literature on mental health apps, including convenience (Carolan and de Visser, 2018) and the personalisation of scalable interventions (Borghouts et al., 2021). However, other insights shared by respondents point to the importance of the relational elements of the intervention, and to group participants' sense of the programme as 'real' (rather than virtual). Difficulties arise, however, because precisely the factors which give the programme its 'real' feel for participants, are those which pose barriers to scale (synchronous delivery, the requirement of a clinician to deliver content, and need for strong Internet bandwidth among users). These contrasting findings also point to the potential benefits of combining multiple forms of digital intervention, such

as augmenting a mobile app with access to group therapy on a digital platform, which could enable further customization.

Finally, the experiences of students remind us that no intervention is likely to meet the needs and preferences of all users. While the online group appealed to many students, there were others who felt that the intervention was not well suited to them. This highlights the importance of person-centred approaches to digital solutions, and the need for including a range of interventions within student counselling centres so students can be matched with those that are potentially most helpful to them.

The implications of the findings of this study for the delivery of digital mental health interventions include:

1. There is a need for psychoeducation among students about different types of digital mental health intervention to set expectations prior to engagement;
2. Ways to increase opportunities for peer engagement in digital mental health interventions need to be identified;
3. Interventions must be developed with in-built mechanisms that support responsiveness to group needs and context;
4. Users need to be supported to minimise distractions during engagement with online mental health programming;
5. Efforts need to be made to expand the range of digital mental health intervention options available to students and allow students to excise autonomy in selecting the one best suited to their needs, and these options should include programmes of low- and high-intensity, and low- and high-digitisation and
6. It will be important for the field to improve understanding of the individual-level factors which predict engagement and treatment response with different types of digital interventions so that prediction algorithms can be developed to personalise triage.

We need to leverage these findings to support the uptake and implementation of various digital mental health resources across university campuses, capitalising on the flexibility of digital offerings to meet the demand for mental health support among students.

Despite the value of these findings, some limitations of the study must be noted. Firstly, only 12 out of 60 randomly recruited participants completed an in-depth interview, indicating some degree of selection bias. However, efforts were also made to assess the degree to which the codes and themes identified in the questionnaire dataset were mirrored in the in-depth interviews, suggesting that many of the major findings resonated across the majority of participants. Secondly, the university at which the study was conducted is significantly better resourced than many other universities in SA, and its student population is not representative of the broader demographics of the country's university population, having a higher proportion of students who are White and from higher SES backgrounds, and a lower proportion of first-generation students than some other universities. Moreover, the sample (both for the intervention itself and the qualitative evaluation) was overwhelmingly female. In SA, this is often the case in voluntary psychosocial and well-being interventions, possibly related to gender norms regarding distress and help-seeking (Atik and Yalçin, 2011; Juvrud and Rennels, 2017). As such, caution should be applied in considering the implications of these findings for digital mental health programming in SA and other LMICs, and efforts must be made to replicate this methodology across a range of different campuses. Specifically, this highlights the need to conduct similar research on the potential for digital mental health interventions in lower-resourced universities in SA to compare and contrast

findings with the current study. Further, this is particularly important given that this study took place during the more acute phases of SA's COVID-19 pandemic, and so it will also be important for future studies to understand how the factors identified in this study play out outside of the pandemic context. Finally, no formal inter-rater agreement reliability statistics were calculated. While disagreements were minimal, and resolved by a senior coder, as noted, this is a limitation of the present analysis.

**Open peer review.** To view the open peer review materials for this article, please visit http://doi.org/10.1017/gmh.2023.39.

**Supplementary material.** The supplementary material for this article can be found at https://doi.org/10.1017/gmh.2023.39.

**Data availability statement.** All transcripts to support the study are available upon request to the authors.

**Acknowledgements.** We would like to acknowledge the hard work of the therapists who implemented the intervention, as well as the students who gave their time to complete the survey and participate in interviews.

**Author contribution.** Analysis: D.C.J., E.B., J.A.N., X.H.; Conceptualisation: J.B.; Editing and revising drafts: All authors; First draft: X.H.; Methodology and data collection: D.C.J., E.B.

**Financial support.** The work reported herein was made possible through funding by the South African Medical Research Council (SAMRC) through its Division of Research Capacity Development under the MCSP (awarded to J.B.) and the National Research Foundation (NRF) (Grant number 142143, awarded to J.B.). The content hereof is the sole responsibility of the authors and does not necessarily represent the official views of the SAMRC or NRF.

**Competing interest.** The authors declare none.

**Ethics statement.** These interviews were recorded, transcribed and analysed. Ethical approval was obtained from the Psychology Ethics Committee, Stellenbosch University (N19/10/145, Project ID: 12977). All participants provided informed consent.

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
