## [Reviewer Report]

Dear Editors

It gives us great pleasure to submit our paper entitled “South African university students’ experiences of online group cognitive behavioural therapy: Implications for delivering digital mental health interventions to young people”. We believe that the piece makes an important contribution to the literature on digital interventions among young people in low- and middle-income countries, as it shows how implementation and contextual factors shape delivery and perceived impact.

Additional highlights of the work include a conceptual contribution to thinking about online group therapy as a modality.

We have no conflicts of interest to disclose.

Thank you for your kind consideration of this piece.

Yours sincerely,

Dr Xanthe Hunt, on behalf of the authors

---

## [Reviewer Report]

This paper sets out to evaluate participants’ experiences of an online group cognitive behavioural intervention conducted over 10 weeks during the Covid-19 pandemic in 2020 in South Africa. The intervention itself addresses an important problem, with high levels of mental disorders among students in South Africa and substantial treatment gaps and resource constraints limiting the uptake of traditional in-person psychotherapy solutions. To understand the participants’ experiences of this intervention the authors collected responses to a qualitative questionnaire (n = 125) and further interviewed 12 participants.

The paper is clearly written and generally follows well-established conventions for this type of study. As such, I only have a few comments for the authors at this point.

Introduction:

The introduction and background provide a solid foundation and framing for the study. I appreciate the framework provided in figure 1 that conceptualises digital interventions as falling along two dimensions: digitisation and intensity. It is perhaps worth 1) returning to this in the discussion and 2) expanding the treatment of this framework. In particular, the authors should consider the necessity for a practitioner, the ability of interventions to scale or not, and the degree to which individuals can choose how and when to engage with the interventions (beyond simply the degree of digitalisation).

Following the description of the framework the authors describe the specific intervention adopted in this study - an online group behavioural therapy programme that took the form of video calls with groups of participants. The motivations for this choice—to capitalise on the cost- and scale-benefits of digital accessibility while not losing the effectiveness of in-person therapy—are sound. Despite this, the initial focus and discussion of apps and app-based interventions does appear somewhat out of place given the specific nature of the intervention in question.

The introduction concludes with the provision of a research objective for the study - “to conduct qualitative interviews with participants from the 2020 pilot study to explore their perspectives of the intervention’s content and delivery, and to provide insights to guide future implementation of the GCBT intervention.” Were there particular guiding research questions or aspects of their experiences that the study aimed to uncover other than merely exploring? Did these guide the construction of the interview guide? If not, what informed the development of the interview guide?

Methods:

The methods are well described and follow well-established norms and conventions for this type of study. To further understand the sample and the intervention itself, the paper would benefit from more information on how participants were originally recruited for the GCBT intervention in the first place. In particular, given that this comes up in the qualitative data, the initial framing of the purpose of the intervention would provide further context and understanding for these findings.

To help the reader further understand how the data collected, assuming that no ethics approval for sharing the actual study data was provided, I recommend including the interview guide as an appendix/in the online supplementary material. Further, the instruments section should motivate the factors that guided the selection of questions and areas to focus on in the interviews.

What software was used to code the interviews and survey responses?

Findings:

The description and presentation of the findings are good. However, the framing of the themes and sub-themes as they appear in the section titles and in table 1 appears to be misaligned with thematic as defined by Braun and Clarke. To me, the current themes read more like domain summaries than themes. Braun and Clarke define a theme as follows:

“A theme captures a common, recurring pattern across a dataset, organised around a central organising concept. A theme tends to describe the different facets of a pattern across the dataset. A subtheme exists ‘underneath’ the umbrella of a theme. It shares the same central organising concept as the theme, but focuses on one notable specific element. “

They then define a domain summary as follows:

“The difference between a theme and a domain summary is a source of frequent confusion in much published TA research. A domain summary is a summary of an area (domain) of the data; for example, a summary of everything the participants said in relation to a particular topic or interview question. Unlike themes, there isn’t anything that unifies the description of what participants said about this topic – there is no underlying concept that ties everything together and organises the analytic observations. In our approach to TA, themes are conceptualised as patterns in the data underpinned by a central concept that organises the analytic observations; this is rather different from a domain summary, and the two should ideally not be confused when using our approach. More simply put, a theme identifies an area of the data and tells the reader something about the shared meaning in it, whereas a domain summary simply summarises participant’s responses relating to a particular topic (so shared topic but not shared meaning). “

Currently the themes are:

- Context of the intervention

- Expectations

- Factors affecting uptake and engagement

- Perception of therapeutic value of the intervention.

These are domains within the data (topics) within which one would expect to find themes that capture shared meaning. I think the actual write-up of the findings is fine but the framing and organisation merits revision.

It is also important to acknowledge the role of the pandemic in the intervention and how this might differ going forward. Many of the benefits only existed due to the context at the time. This perhaps limits transferability of some of these findings to other times and contexts.

Similarly, themes around expectations will relate specifically to how the GCBT intervention was framed during the initial recruitment and advertising phases. As I noted above, more information on this is needed to provide the necessary context for these findings.

Overall Evaluation

Overall, this is an interesting and timely study that provides valuable insights into students’ experiences with online video-conferencing based cognitive behavioural interventions. My comments are generally fairly high level and primarily relate to the framing of the study and the organisation of the themes. In terms of implications, in addition to the six outline, I think it is also necessary to understand and manage the expectations of participants in any future such interventions.

---

## [Reviewer Report]

**Summary**

This paper presents a qualitative study to understand university students’ experience with an online group cognitive behavioral therapy intervention. While students overall had a positive experience, there were barriers to engagement related to the digital and group format of the therapy sessions.

The manuscript presents interesting findings and is overall straightforward to read. I have listed my suggestions for improvement below, which I believe can be addressed in a revision of the paper.

**Comments**

• L19: Spell out LMIC when it is first used in the impact statement.

• L125-126: I assume that the intervention is described in more detail in the other publication, but it would be helpful to provide a bit more detail on the intervention here to better contextualize the paper findings. For example, what efforts were made to ensure the intervention does “not lose the effectiveness and acceptability of in-person therapy” and how was it “found to show promise as an effective and sustainable intervention”?

• L149: how were students recruited? Was everyone from the university sent an email, was it promoted during a specific class, was it promoted on social media, etc.? How many students were reached and how many enrolled in the study?

• It is not clear how participants were randomly selected for the interviews. The Method section reads as if a random sample of 12 participants was selected, but the Discussion section implies that 60 participants were selected and that out of these, 12 responded to participate. A more detailed explanation would help here.

• L149: spell out SA when it is first used.

• L164: was one email sent to a different group of 12 participants 5 times, thus resulting in a group of 60? Were people sent reminder emails?

• Did participants have any prior experience with therapy? Participants with no mental health challenges or prior experience with therapy may have a different experience.

• L202-204: were any reliability checks done between coders?

• One of the interview themes related to people having low initial expectations. Is there any data available related to people’s motivations to participate, if they had low expectations of the intervention?

• I would move study limitations to the end of the Discussion, as it reduces the strength and impact of the paper by opening the Discussion with this.

• L455: do the authors have any suggestions on how to increase male participation in future studies?

• The paper presents manualization as a characteristic of the therapy being a digital format, but aren’t in-person CBT and group therapies often manualized as well?

• The paper uses quite a bit of abbreviations and it would be helpful to have a list of the abbreviations and what each stands for at the end of the paper.

---

## [Reviewer Report]

Thank you for submitting your manuscript for review. The reviewers have provided valuable comments and feedback.

We invite you to address the comments and suggestions in your response.

---

## [Reviewer Report]

Dear Editor and Reviewers

Thank you very much for the opportunity to revise our manuscript. We found the reviewers’ feedback helpful, and hope that you find the manuscript strengthened. We would be happy to make any further revisions if needed.

Our findings are detailed in the table below.

Yours sincerely,

Xanthe Hunt, on behalf of the authors

General

Please include the abstract in the main text document.

This was included in the original submission. We are not sure why it was not visible to the reader. However, these sections are reflecting on our proofs on the online ‘view submission’ document, too.

Please include an Impact Statement below the abstract (max. 300 words). This must not be a repetition of the abstract but a plain worded summary of the wider impact of the article.

This was included in the original submission. We are not sure why it was not visible to the reader. However, these sections are reflecting on our proofs on the online ‘view submission’ document, too. We have ensured that the Impact Statement so that it aligns with the journal’s specifications.

Submission of graphical abstracts is encouraged for all articles to help promote their impact online. A Graphical Abstract is a single image that summarises the main findings of a paper, allowing readers to quickly gain an overview and understanding of your work. Ideally, the graphical abstract should be created independently of the figures already in the paper, but it could include a (simplified version of) an existing figure or a combination thereof. If you do not wish to include a graphical abstract please let me know.

We have included a graphical abstract in the submission.

Please ensure references are correctly formatted. In text citations should follow the author and year style. When an article cited has three or more authors the style ‘Smith et al. 2013’ should be used on all occasions. At the end of the article, references should first be listed alphabetically, with a full title of each article, and the first and last pages. Journal titles should be given in full.

Statements of the following are required in the main text document at the end of all articles: ‘Author Contribution Statement’, ‘Financial Support’, ‘Conflict of Interest Statement’, ‘Ethics statement’ (if appropriate), ‘Data Availability Statement’. Please see the author guidelines for further information.

Thank you – these were also submitted on the original document. We hope that they are visible to the reader.

Reviewer 1

This paper sets out to evaluate participants’ experiences of an online group cognitive behavioural intervention conducted over 10 weeks during the Covid-19 pandemic in 2020 in South Africa. The intervention itself addresses an important problem, with high levels of mental disorders among students in South Africa and substantial treatment gaps and resource constraints limiting the uptake of traditional in-person psychotherapy solutions. To understand the participants’ experiences of this intervention the authors collected responses to a qualitative questionnaire (n = 125) and further interviewed 12 participants.

The paper is clearly written and generally follows well-established conventions for this type of study. As such, I only have a few comments for the authors at this point.

Thank you very much for your feedback on our paper, and the constructive comments. We hope that you find the submission strengthened following our revision of it.

Introduction:

The introduction and background provide a solid foundation and framing for the study. I appreciate the framework provided in figure 1 that conceptualises digital interventions as falling along two dimensions: digitisation and intensity. It is perhaps worth 1) returning to this in the discussion and 2) expanding the treatment of this framework. In particular, the authors should consider the necessity for a practitioner, the ability of interventions to scale or not, and the degree to which individuals can choose how and when to engage with the interventions (beyond simply the degree of digitalisation).

Thank you for this reflection. We have added a section to the discussion to draw this link – between the findings and this original framing – more clearly.

Following the description of the framework the authors describe the specific intervention adopted in this study - an online group behavioural therapy programme that took the form of video calls with groups of participants. The motivations for this choice—to capitalise on the cost- and scale-benefits of digital accessibility while not losing the effectiveness of in-person therapy—are sound. Despite this, the initial focus and discussion of apps and app-based interventions does appear somewhat out of place given the specific nature of the intervention in question.

Thank you for drawing our attention to this. We have refined and reduced some of the discussion of apps in the background section, and increased the discussion of online therapies, so that the specific intervention described is better aligned with the background.

The introduction concludes with the provision of a research objective for the study - “to conduct qualitative interviews with participants from the 2020 pilot study to explore their perspectives of the intervention’s content and delivery, and to provide insights to guide future implementation of the GCBT intervention.” Were there particular guiding research questions or aspects of their experiences that the study aimed to uncover other than merely exploring? Did these guide the construction of the interview guide? If not, what informed the development of the interview guide?

We have added detail on this to section 2.4. Please see the section which starts as follows, These questions were guided by our desire to understand the following…

Methods:

The methods are well described and follow well-established norms and conventions for this type of study. To further understand the sample and the intervention itself, the paper would benefit from more information on how participants were originally recruited for the GCBT intervention in the first place. In particular, given that this comes up in the qualitative data, the initial framing of the purpose of the intervention would provide further context and understanding for these findings.

Thank you for raising this. We have added some information about the recruitment for the intervention study, to the methods section. See the section starting, Recruitment for the pilot open label trial is…

To help the reader further understand how the data collected, assuming that no ethics approval for sharing the actual study data was provided, I recommend including the interview guide as an appendix/in the online supplementary material. Further, the instruments section should motivate the factors that guided the selection of questions and areas to focus on in the interviews. We have included the interview schedule as a Supplementary File and provided an explanation of the choice of focus areas for the questionnaire under 2.4 Instruments.

What software was used to code the interviews and survey responses?

We used Microsoft Excel to code the transcripts. We have added detail on this process under 2.6 Data Analysis.

Findings:

The description and presentation of the findings are good. However, the framing of the themes and sub-themes as they appear in the section titles and in table 1 appears to be misaligned with thematic as defined by Braun and Clarke. To me, the current themes read more like domain summaries than themes. Braun and Clarke define a theme as follows:

“A theme captures a common, recurring pattern across a dataset, organised around a central organising concept. A theme tends to describe the different facets of a pattern across the dataset. A subtheme exists ‘underneath’ the umbrella of a theme. It shares the same central organising concept as the theme, but focuses on one notable specific element. “

They then define a domain summary as follows:

“The difference between a theme and a domain summary is a source of frequent confusion in much published TA research. A domain summary is a summary of an area (domain) of the data; for example, a summary of everything the participants said in relation to a particular topic or interview question. Unlike themes, there isn’t anything that unifies the description of what participants said about this topic – there is no underlying concept that ties everything together and organises the analytic observations. In our approach to TA, themes are conceptualised as patterns in the data underpinned by a central concept that organises the analytic observations; this is rather different from a domain summary, and the two should ideally not be confused when using our approach. More simply put, a theme identifies an area of the data and tells the reader something about the shared meaning in it, whereas a domain summary simply summarises participant’s responses relating to a particular topic (so shared topic but not shared meaning).

Thank you for providing this detailed description and these recommendation for presentation of our findings. Please see our specific responses to your comments below.

Currently the themes are:

- Context of the intervention

- Expectations

- Factors affecting uptake and engagement

- Perception of therapeutic value of the intervention.

These are domains within the data (topics) within which one would expect to find themes that capture shared meaning. I think the actual write-up of the findings is fine but the framing and organisation merits revision.

Thank you very much for this useful reflection. We have renamed the themes to reflect, more accurately, what is ‘going on’ within the domain. So, the theme titles have moved from being a designation of the area of information covered and are now more descriptive of the findings encompassed in the themes. These new theme names are:

• Opportunities for connection and continuity

• Reality versus expectations

• Group format and online spaces as barriers and facilitators

• Perception of therapeutic value of the intervention

It is also important to acknowledge the role of the pandemic in the intervention and how this might differ going forward. Many of the benefits only existed due to the context at the time. This perhaps limits transferability of some of these findings to other times and contexts.

This is an important caveat to our findings, and we have mentioned it in the limitations section, now (see the paragraph starting ‘Before proceeding with an in-depth…”

Similarly, themes around expectations will relate specifically to how the GCBT intervention was framed during the initial recruitment and advertising phases. As I noted above, more information on this is needed to provide the necessary context for these findings. We have revised the section on expectations to more explicitly discuss how the intervention was initially framed to the participants.

We have also added detail on the original recruitment process, to the methods section (see above).

Overall, this is an interesting and timely study that provides valuable insights into students’ experiences with online video-conferencing based cognitive behavioural interventions. My comments are generally fairly high level and primarily relate to the framing of the study and the organisation of the themes. In terms of implications, in addition to the six outline, I think it is also necessary to understand and manage the expectations of participants in any future such interventions.

Thank you so much for your comments on the paper – they were extremely helpful. We have also added some more information, in the limitations section, regarding how managing expectations and appropriate advertising/recruitment can be managed in the context of digital interventions like this one.

Reviewer 2

This paper presents a qualitative study to understand university students’ experience with an online group cognitive behavioural therapy intervention. While students overall had a positive experience, there were barriers to engagement related to the digital and group format of the therapy sessions. The manuscript presents interesting findings and is overall straightforward to read. I have listed my suggestions for improvement below, which I believe can be addressed in a revision of the paper.

Thank you so much for your helpful comments. We hope that you find the revised paper strengthened.

L19: Spell out LMIC when it is first used in the impact statement.

Amended in text.

L125-126: I assume that the intervention is described in more detail in the other publication, but it would be helpful to provide a bit more detail on the intervention here to better contextualize the paper findings. For example, what efforts were made to ensure the intervention does “not lose the effectiveness and acceptability of in-person therapy” and how was it “found to show promise as an effective and sustainable intervention”?

We have added this detail to the methods section.

L149: how were students recruited? Was everyone from the university sent an email, was it promoted during a specific class, was it promoted on social media, etc.? How many students were reached and how many enrolled in the study?

Thank you for raising this point. We have added a detailed piece on recruitment to the section 2.3 Participants and Procedure. 175 students were enrolled in the pilot RCT from which the qualitative sub-sample for this analysis were taken.

It is not clear how participants were randomly selected for the interviews. The Method section reads as if a random sample of 12 participants was selected, but the Discussion section implies that 60 participants were selected and that out of these, 12 responded to participate. A more detailed explanation would help here.

We have significantly revised the section 2.3 Participants and Procedure to provide greater clarity on the process.

L149: spell out SA when it is first used.

Amended in text.

L164: was one email sent to a different group of 12 participants 5 times, thus resulting in a group of 60? Were people sent reminder emails?

This detail has also been added to section 2.3:

Five rounds of recruitment emails were sent to 12 participants each time (60 in total). In each recruitment batch, the 12 participant email addresses were randomly selected from the total sample using a random number generator. One reminder email was sent to each participant before a new batch of emails was sent out. The process of sending recruitment emails was continued until the sample size for the in-depth interviews had been achieved (n=12).

Did participants have any prior experience with therapy? Participants with no mental health challenges or prior experience with therapy may have a different experience.

This is an important point. We have added context on the recruitment, as noted above, and in this section, we have specified that participants did not need to have any symptoms in order to participate (there were no symptom thresholds for participation).

L202-204: were any reliability checks done between coders?

No quantitative reliability checks were conducted – so no inter-rater agreement statistics can be reported (and this is not always recommended for qualitative work). This has been added to the limitations section. See the section starting Finally, because coding was conducted in Microsoft Excel, no formal inter-rater agreement…

One of the interview themes related to people having low initial expectations. Is there any data available related to people’s motivations to participate, if they had low expectations of the intervention?

Unfortunately, we do not have data on motivations to participate (although this is something we would like to explore in future work). Our sense, however, regarding the expectation-setting piece, is that this particular group of students assumed ‘psychological skills training’ to be more like a course, than like therapy. Perhaps this is in part due to the fact that it was delivered on a university campus, and so the context cues are heavily weighted towards educational offerings rather than therapeutic ones.

I would move study limitations to the end of the Discussion, as it reduces the strength and impact of the paper by opening the Discussion with this.

Amended in text, thank you.

L455: do the authors have any suggestions on how to increase male participation in future studies?

This is the million-dollar question. One option, which we have now mentioned in text, is that specific social marketing strategies need to be tested and their effectiveness evaluated using gender-disaggregated data.

The paper presents manualization as a characteristic of the therapy being a digital format, but aren’t in-person CBT and group therapies often manualized as well?

This is a good point. Many in-person CBT groups are manualised, too. However, with digitisation comes immense opportunities to scale. And it is this scaling quickly and to massive proportions that means that manualisation risks becoming reductive. Where an intervention – say the in-person versions of Thinking Healthy – is manualised but training is still done at a relatively small scale, there are opportunities to refine delivery for audience, and by context, setting etc. Whereas, with the opportunities afforded by digital platforms, these opportunities may be lacking.

The paper uses quite a bit of abbreviations and it would be helpful to have a list of the abbreviations and what each stands for at the end of the paper.

We are asking the journal if they permit a list of abbreviations to be included at the start of the article. If they do, we will certainly make one.

---

## [Reviewer Report]

I am satisfied with the revisions made in response to my comments and am happy to recommend acceptance.

---

## [Reviewer Report]

I thank the authors for their responses to reviewer comments and revision of the paper.

I am happy with the changes made and suggest acceptance of the paper.

---

## [Reviewer Report]

Thank you for responding to the reviewers comments, suggestions and submitting an updated manuscript. Both reviewers are satisfied with your response and the updated manuscript and have recommended that we accept your manuscript.